# Impact of Economic Development Level and Agricultural Water Use on Agricultural Production Scale in China

**DOI:** 10.3390/ijerph18179085

**Published:** 2021-08-28

**Authors:** Jiaxing Pang, Ningfei Wang, Xue Li, Xiang Li, Huiyu Wang, Xingpeng Chen

**Affiliations:** 1College of Earth and Environmental Sciences, Lanzhou University, Lanzhou 730000, China; wangnf18@lzu.edu.cn (N.W.); lx20@lzu.edu.cn (X.L.); lixiang2020@lzu.cn (X.L.); chenxp@lzu.edu.cn (X.C.); 2Institute of County Economic Development, Lanzhou University, Lanzhou 730000, China; 3College of Geology and Jewelry, Lanzhou Resources and Environment Voc-Tech College, Lanzhou 730000, China; Wanghuiyu0709@163.com

**Keywords:** the scale of agricultural production, economic development level, agricultural water use, urbanization level, the market price of agricultural products

## Abstract

The change of agricultural production scale is directly related to food security and the stable development of social economy. Particularly, the influence of economic development level and agricultural water use on agricultural production scale cannot be ignored. Therefore, this paper uses the fully modified ordinary least squares (*FMOLS*) and the Dumitrescu–Hurlin panel causality test models to discuss the effects of the level of economic development, agricultural water use, the level of urbanization, and the market price of agricultural products on the scale of agricultural production in China. The analysis results indicated that agricultural water use, the level of urbanization, and the market price of agricultural products promoted an increase of the scale of agricultural production at the total sample level; a 1% increase for these three variables will result in an increase of the scale of agricultural production of 0.634%, 0.377%, and 0.292%, respectively. The influence of economic development level on agricultural production scale accords with Kuznets curve. However, at the regional level, the influence of each variable on the eastern region is consistent with the trend of the total sample. In the central region, the impact of economic development on agricultural production scale shows a U-shaped curve, and the improvement of urbanization level inhibits the expansion of agricultural production scale. In the western region, all variables failed to pass the significance test. The results of the FMOLS model were validated by the fixed effects model. The results of causality tests showed that bidirectional causality existed between the scale of agricultural production and the level of economic development, the scale of agricultural production and agricultural water use, the level of economic development and the market price of agricultural products, and the level of urbanization and the market price of agricultural products. In different regions, there were differences in causality between variables. Therefore, based on the empirical results, we put forward some policy suggestions to maintain the scale of agricultural production.

## 1. Introduction

Water and cultivated land resources are significant contributors to food security and sustainability of the ecosystem. Cultivated land is the basic resource and material guarantee that human beings depend on for survival and development [1,2], and provides food guarantee for the development of human society [3]. Since ancient times, food has been a key issue in maintaining social stability. Therefore, cultivated land plays an important role in global and national food and ecological security [4]. However, due to the rapid growth of population, urban expansion, water shortage, and the impact of climate change, the cultivated land resources in some areas are rapidly decreasing [5,6,7]. The protection and improvement in the quantity and quality of cultivated land cannot be delayed; therefore, a large number of countries have formulated corresponding cultivated land protection policies to ensure the safety of food production [2,8,9].

China has been a big agricultural country since ancient times, and grain output has always been the most basic problem of national livelihood, while the change of cultivated land quantity in time and space has a significant impact on grain output [4]. With the increase of our population, the area of cultivated land has been under threat [2]. China’s total amount of cultivated land resource ranks fourth in the world [10], although its population ranks first in the world. Per capita cultivated land is less than half of the world average [11], and with China facing enormous pressure on cultivated land, therefore, the Chinese government has formulated strict farmland protection policies, such as the basic farmland protection system. In 2013, the Central Rural Conference stated that China would adhere to the red line of 1.8 billion mu of arable land and complete the delineation of permanent basic farmland in 2018. The delimitation of 1.8 billion mu of cultivated land red line and 1.55 billion mu of permanent basic farmland in China has important practical significance to ensure the stability of arable land area to the maximum extent.

China is the world’s second largest economy and has made great achievements in economic development. With the development of economic level, the degree of cultivated land used for non-agricultural purposes is increasing [10], which will further affect the change of agricultural production scale. In recent years, there are still acts of illegally occupying cultivated land for non-agricultural construction, some illegally occupying permanent basic farmland for afforestation, and some illegally occupying cultivated land for over-standard green belt construction on both sides of high-speed railway, national and provincial roads, and rivers, which pose a threat to national food security. In 2019, China’s grain sown area was 1.741 billion mu, more than 40 million mu less than in 2016. In 2020, The General Office of the State Council issued a circular on resolutely stopping the “non-agricultural” conversion of cultivated land. With the rapid urbanization and the continuous expansion of the city scale in China, cultivated land area has been under threat, and the pressure to protect cultivated land is higher and higher. As the most populous country in the world, the contradiction between population and cultivated land is particularly prominent in China, accounting for 22% of the global population with only 7% of the world’s cultivated lands [12,13]. In recent years, due to the increasing demand for urban construction land, the area of cultivated land has been decreasing in China, while the urban population has been increasing. Relevant studies showed that China lost 4.73 million hectares of cultivated land from 1978 to 1996, mainly in the eastern coastal and central provinces [14]. According to the statistics, the urban population of China was 435.04 million in 2019, an increase of 139.14 million compared with 295.89 million in 1991; the urban built-up area of China reached 6.03 million hectares in 2019, an increase of 4.63 million hectares compared with 1.40 million hectares in 1991. In 2019, 0.27 million hectares of land were requisitioned for urban construction, of which 0.12 million hectares were cultivated land; the urban construction requisitioned 1.54 million hectares of cultivated land from 2000 to 2019. In China, the urbanization level rose from 17.92% in 1978 to 60.60% in 2019, and will rise by 70% in 2030 [15]. The direct effects resulted from the urbanization process led to a large amount of land converted to construction land and the migration of a large number of rural population to cities [16]. Urbanization leads to the separation of labor and cultivated land, and the emergence of “concurrent farming” or “abandonment of agriculture”, which leads to the abandonment of cultivated land [17,18,19]. Based on a sample survey, the researchers found that the rate of cultivated land abandonment in mountainous areas of China was 14.32% from 2014 to 2015, the abandonment rate was the highest in the mountainous areas along the Yangtze River Basin [20]; at the same time, some scholars used meta-analysis to reveal the spatial and temporal characteristics of cultivated land abandoned at the national scale in China, and found that the abandoned land was mainly concentrated in southern China [21].

Agricultural production is closely related to water resources, but the distribution of water resources is uneven in China [22]. Water resources play an important role in agriculture and directly affect agricultural production [23]. The matching degree of cultivated land and water resources is the basic premise of agricultural efficient production [24], therefore, the distribution of water resources may affect the scale of regional agricultural production. Agriculture is the largest user of water, accounting for about 70% of global freshwater withdrawals [25,26]. The current distribution of crops around the world is neither attaining maximum production nor minimum water usage [27]. To maximize crop yields, a 146% increase in global irrigation water is needed [26]. Cultivated land and water resources are two rigid constraints affecting grain production, the use of cultivated land and the use of water resources are interrelated, the utilization of water resources will affect the expansion of agricultural production scale. At present, the total supply and demand of grain is basically balanced in China, but we need to be seriously aware that the problems of water resources and cultivated land are only temporarily alleviated, and long-term problems still exist.

In the process of production, resources interact with each other, and the degree of interaction is becoming more and more significant. The concept of the two-sector nexus as water-food, water-land, land-food, water-energy, energy-food, and land-energy are common among researchers [28]. With cultivated land resources as the basic carrier, it is particularly important to study the influence of water resources, economic development level, product market price, and other factors on cultivated land. Therefore, based on China’s provincial panel data, this paper focuses on the effects of water resources, economic development level, product market price, and urbanization level on China’s agricultural production scale. It is hoped that this study will be of some help to the stability of China’s agricultural production scale.

## 2. Data and Methodology

### 2.1. Data

According to the availability of the variables, 30 provinces, except Tibet, Taiwan, Hong Kong, and Macao, were selected as the research objects, and the study period was from 2003 to 2019. The relevant data used in this study comes from the China Statistical Yearbook (2004–2020), and the China Rural Statistical Yearbook (2004–2020), and provincial statistical data. The scale of agricultural production is expressed by the sown area of crops (AS); the level of economic development is expressed by GDP per capita (EG); and agricultural water use is expressed in terms of total agricultural water use (AW). At the same time, the urbanization level and the market price of agricultural products are introduced as control variables. The level of urbanization is expressed as the urbanization rate (UR); the market price of agricultural products is expressed by the production price index of agricultural products (PFI). Per capita GDP and the producer price index of agricultural products are converted to 2003 as the base year.

In order to analyze the interrelationship among various elements in different regions, the eastern, central, and western regions are divided. The eastern region includes Beijing, Tianjin, Hebei, Liaoning, Shanghai, Jiangsu, Zhejiang, Fujian, Shandong, Guangdong, Guangxi, and Hainan; the central region includes Shanxi, Inner Mongolia, Jilin, Heilongjiang, Anhui, Jiangxi, Henan, Hubei, and Hunan; the western region includes Chongqing, Sichuan, Guizhou, Yunnan, Shaanxi, Gansu, Qinghai, Ningxia, and Xinjiang.

### 2.2. Model Specification

In order to study the long-term relationship between the level of economic development, agricultural water consumption, and sown area in China and the eastern, central, and western regions from 2003 to 2019, we introduced control variables, at the same time, by introducing the square of economic development level, this paper tries to verify the existence of Kuznets curve characteristics between economic development level and agricultural production scale. The specific model followed was adopted:$$AS=f(EG,EG^{2},AW,UR,PFI)$$

In order to reduce the issues of heteroscedasticity, we took the natural logarithm of the formula; the formula can be written in the following form:$$\ln AS_{it}=\alpha_{i}+\beta_{1}\ln EG_{it}+\beta_{2}\ln EG^{2}+\beta_{3}\ln AW_{it}+\beta_{4}\ln UR_{it}+\beta_{5}\ln PFI_{it}+\varepsilon_{it}$$
where β is coefficient, ε is error term.

### 2.3. Panel Unit Root Test

We needed to make sure that the series were stationary before we analyzed the data, otherwise, it would be possible to produce spurious regression. The panel unit root test was used to examine the stationary of all variables. The study used the Levin–Lin–Chu (LLC) and Im–Pesaran–Shin (IPS) unit root tests [29,30].

### 2.4. Panel Cointegration

The Johansen cointegration test is a test to see the long-term equilibrium relationship among variables one to the others. The Johansen cointegration test derives two tests, the trace test, and the maximum eigenvalue test to check cointegration. Each test has slightly different criterion [31]. 

The trace test and the maximum eigenvalue test are represented as:$$\lambda_{trace}=-T\sum_{i=r+1}^{n}\ln(1-\lambda_{i})$$
$$\lambda_{\max}=-T\ln(1-\lambda_{r+1})$$
where *T* is the sample size, $\lambda_{i}$ is the *i*th largest canonical correlation. The trace statistic tests the null hypothesis of r cointegrating relations against the alternative of *r* + 1 cointegrating relations. The maximum eigenvalue tests the null of r cointegrating vectors against the alternative hypothesis of *r* + 1 cointegrating vectors. Determining the optimal lag order: firstly, the VAR model was established; secondly, the optimal lag order is determined according to the VAR model.

### 2.5. FMOLS

In the real complex economic activities, the input elements of production activities are closely linked, and the macroeconomic fluctuations will inevitably lead to the change of the activities of the elements, so it is difficult to avoid the endogenous and serial correlation problems of the input elements. The analysis of long-run cointegrating relationships has received considerable attention in modern-time series analysis. Therefore, we chose the fully modified ordinary least squares (*FMOLS*) model. The *FMOLS* was developed by Phillips and Hansen in order to administer an optimal cointegrating regression estimation [32]. The *FMOLS* test takes into account the defects of endogeneity and serial correlation, and the results of the analysis are meaningful and robust. Therefore, the panel *FMOLS* estimator optimized by Pedroni was used in this study [33]. The formula is as follows:$$\beta_{FMOLS}=\frac{1}{N}\sum_{i=1}^{N}\left[\sum_{t=1}^{T}{\left(X_{it}-X_{i}\right)}^{2}\right]\left[\sum_{i=1}^{T}\left(X_{it}-X_{i}\right)Y_{it}^{*}-T\tau_{i}\right]$$
where the necessary analysis is made, assuming that the related *t*-statistic is normally distributed. The results were tested for significance by *t* statistics and *p* values.

### 2.6. Granger Causality Test

To analyze the panel causality between variables, the Dumitrescu–Hurlin [34] panel causality methods are applied in this paper. The formula is as follows [34,35]:$$Y_{it}=\alpha_{i}+\sum_{k=1}^{K}\gamma_{ik}Y_{i,t-k}+\sum_{k=1}^{K}\beta_{ik}X_{i,t-k}+\varepsilon_{it}$$
where *X_it_* and *Y_it_* are the observations of two stationary variables. The lag order is the same for all individuals, and the panel data must be stable.

## 3. Results and Discussion

### 3.1. Data Inspection

Before the results of the unit root test were obtained, it was necessary to analyze the cross-sectional dependence of the variables. We used the Breusch–Pagan LM test, bias-corrected scaled LM, and Pesaran test to check the cross-sectional dependence of the variables. We found that the cross-sectional dependence was rejected at the significance level of 1%; the results of the cross-section dependence test are shown in Table 1. This implies there was a presence of cross-sectional dependence in the underlying data. This indicates that the variation of variables between regions can influence each other. Thus, in the case of cross-sectional dependence, the panel unit root test is used to test the data stability.

To avoid false regression, the study employed the Levin–Lin–Chu (LLC) unit root test and Im–Pesaran–Shin (IPS) unit root test to check unit root test of panel data. The results of the panel unit root test are shown in Table 2. The results showed that all variables rejected the null hypothesis of unit root for the entire panel when we applied these tests to the first difference data series. The results indicated that the variables were nonstationary at levels and stationary at first differences. Thus, the study proceeded to examine the presence of cointegration among the AS, EG, AW, UR, and PFI.

All variables are integrated and first order, we researched the long-run cointegrated relationship among the AS, EG, AW, UR, and PFI using the analysis of the Fisher-type Johansen panel cointegration test. The results of the Johansen cointegration test are shown in Table 3. According to the trace statistic and maximum Eigen statistic, results found that the null hypothesis of having no cointegration was rejected. Thus, there are at least three cointegration relationships between all variables. These results confirm the significant long-run equilibrating relationship among the concerned variables.

### 3.2. FMOLS Estimate

Table 3 shows that there was a long-run relationship among the variables; however, it does not show how the independent variables affect the dependent variables and whether it has a negative or positive effect. Therefore, we selected the FMOLS estimator to determine the long-run relationships among the variables. The long-run results of the FMOLS estimation are shown in Table 4. According to the results in Table 4, we find that agricultural water consumption has an important positive impact on the scale of agricultural production, but the influence of economic development level on agricultural production scale is not consistent in different regions. 

At the national level, the impact of economic development level on agricultural production scale is not significant at the 10% significant level. However, the square of economic development level inhibited the expansion of agricultural production scale at the significant level of 5%, if the square of economic development level increased by 1%, the agricultural production scale decreased by 0.026%—this trend is consistent with the Kuznets curve. Thus, the agricultural production scale does not depend on the economic development level, but it is largely affected by other variables. Agricultural water consumption, urbanization rate, and agricultural production price index all promoted the scale of agricultural production at the significant level of 1%, the corresponding indices are 0.634, 0.377, and 0.292, respectively. 

In the eastern region, the influence pattern of the level of economic development and the square of the level of economic development on the scale of agricultural production was consistent with that of the national level. The level of economic development does not pass the 10% significance test, while the square of the level of economic development passes the 10% significance test; the square of economic development level increased by 1%, the agricultural production scale decreased by 0.030%. Agricultural water consumption, urbanization rate, and agricultural production price index all promoted the scale of agricultural production at the significant level of 1%; the corresponding indices were 0.836, 0.613, and 0.602, respectively. In the central and western regions, the influence of the level of economic development and the square of the level of economic development on the scale of agricultural production was completely opposite to the national level and the eastern region, which showed the characteristics of a U-shaped curve. In the central region, the level of economic development inhibits the expansion of agricultural production scale, while the square of economic development level promoted the expansion of agricultural production scale; the level of economic development increased by 1%, the scale of agricultural production decreased by 1.678%; the square of the level of economic development increased by 1%, the scale of agricultural production increased by 0.085%, and they all passed the significance test of 1%. Agricultural water consumption promoted the expansion of agricultural production scale and passed the significance test of 10%; producer price index promoted the expansion of agricultural production scale and passed the significance test of 1%; the urbanization rate inhibited the expansion of agricultural production scale and passed the significance test of 1%. The central region is one of the major grain-producing areas in China. With the improvement of urbanization rate, the rural population continues to migrate to cities, which leads to a decrease of agricultural workers and results in part of the arable land become abandoned, thus affecting the decrease of the sown area, which is consistent with the existing research results [21]. In the western region, the effect of all variables on the scale of agricultural production does not pass the 10% significance test. The western region is relatively short of cultivated land resources, and most of them are dry farming production mode. At present, the overall supply of cultivated land is insufficient, so the influence of each variable on the scale of agricultural production is not significant. Thus, the scale of agricultural production in the western region is greatly impacted by other variables. 

The effect of economic development level on agricultural production scale is different in different regions. As a region with a high level of economic development in China, the eastern region conforms to the characteristics of Kuznets curve. At a relatively low level of economic development, it depends on agricultural production to ensure food supply; when the economic level was high, some agricultural land was converted into construction land, people could obtain higher income through other sectors, and some agricultural practitioners could obtain higher income through other industries, thus reducing the scale of agricultural production. However, the eastern and western regions are important grain production areas in China, and the arable land area accounts for about 75% of the total arable land area in China; as the level of economic development increases, local governments will invest more resources to ensure that the scale of agricultural production increases. As an important factor of agricultural production, water resource plays a positive role in promoting the expansion of agricultural production scale. Agricultural irrigation water occupies a large amount of water resources, but the effective utilization coefficient of irrigation water is relatively low, because there is still a large amount of waste in the process of irrigation transportation. Therefore, agricultural water efficiency needs to be improved to ensure the scale of agricultural production. Producer price index of agricultural products reflected the changing trend of the selling price of agricultural products; the producer price index of agricultural products promotes the expansion of agricultural production scale, which is in line with the principle of supply. The urbanization rate inhibited the expansion of agricultural production scale in the areas with rich arable land resources and promoted the agricultural production scale in the areas with poor arable land resources. This is because in the areas with rich arable land, the workers have reached the scale of marginal work, so the urbanization rate reduces agricultural practitioners, thereby inhibiting the scale of agricultural production; for the areas with insufficient arable land resources, the workers do not reach the marginal work scale, and the increase of urbanization rate will increase the cultivable scale of practitioners.

To ensure the reliability of the results, we used the fixed-effect model to test robustness of the results of the panel FMOLS model. The results are shown in Table 5. We find that the fixed-effect model and the FMOLS model produce consistent long-run trend results, the fixed-effect model validates the results of the FMOLS model, and therefore, we think that the fixed-effect method provides robust and reliable verification on the results of the FMOLS model.

### 3.3. Panel Causality Test

The Johansen cointegration test confirmed the existence of cointegration relationship between variables. Cointegration analysis of the results obtained only long-run equilibrium relationship between variables, and not necessarily a causal relationship between them; the causal relationship between the variables needed to be Granger causality tested to verify. Therefore, we further used the pairwise Dumitrescu–Hurlin panel and Granger causality tests to test the causality between the variables. The results are shown in the Table 6 and Figure 1.

For the total sample, we found that there existed bidirectional causality between the scale of agricultural production and the level of economic development, the scale of agricultural production and agricultural water use, the level of economic development and the market price of agricultural products, and the level of urbanization and the market price of agricultural products. There is unidirectional causality between the market price of agricultural products and the scale of agricultural production, the level of urbanization and the scale of agricultural production, the level of economic development and agricultural water use, the level of economic development and the level of urbanization, the market price of agricultural products and agricultural water use, and the level of urbanization and agricultural water use.

For the region sample, we found that the causality of variables in the eastern region is similar to that in the western region, the main difference between the two regions is that there is bidirectional causality between the market price of agricultural products and the level of urbanization, independent causality between the scale of agricultural production and the level of urbanization in the eastern; however, there is unidirectional causality between the level of urbanization and the scale of agricultural production, the market price of agricultural products and the level of urbanization in the western region.

## 4. Conclusions

With the development of the economy, the intensification of agricultural water use, the improvement of the urbanization, the development of the market economy, and the change in the amount of cultivated land, better researching on the effects of the level of economic development, agricultural water use, the level of urbanization, and the market price of agricultural products on the scale of agricultural production is conducive to the implementation of farmland protection and the guarantee of food security. This study utilizes the FMOLS model to estimate long-run equilibrium relationships between variables, and pairwise Dumitrescu–Hurlin panel causality tests were used to study the causal relationship between variables.

Some of the important finds were shown as follows: where other factors remained unchanged, at the significance level of 10%, the panel FMOLS test results of the total sample indicated that agricultural water use, the level of urbanization, and the market price of agricultural products promoted the increase of the scale of agricultural production; a 1% increase for these three variables will result in an increase of the scale of agricultural production of 0.634%, 0.377%, and 0.292%, respectively. The economic development level and agricultural production scale showed an inverted U—shaped relationship, but the level of economic development did not pass the significance test. At the regional sample, agricultural water consumption has an important impact on the scale of agricultural production, and the influence of economic development level on agricultural production scale was not consistent in different regions. 

At the significance level of 10%, the panel FMOLS test results of the regional sample indicated that the influence of each variable on the eastern region was consistent with the trend of the total sample. In the central region, the effects of all variables on agricultural production scale passed the significance test of 10%, the impact of economic development on agricultural production scale showed a U-shaped curve, and the improvement of urbanization level inhibited the expansion of agricultural production scale; agricultural water use and the market price of agricultural products would lead to the increase of agricultural production scale. In the western region, except that the level of economic development inhibits the increase of agricultural production scale, other variables promote the increase of agricultural production scale, but all of them failed to pass the significance test of 10%. For the full sample, the results of causality tests showed that the scale of agricultural production and the level of economic development, the scale of agricultural production and agricultural water use, the level of economic development and the market price of agricultural products, the level of urbanization and the market price of agricultural products, there existed bidirectional causality between them. There is unidirectional causality between the market price of agricultural products and the scale of agricultural production, the level of urbanization and the scale of agricultural production, the level of economic development and agricultural water use, the level of economic development and the level of urbanization, the market price of agricultural products and agricultural water use, and the level of urbanization and agricultural water use.

Therefore, based on the empirical results, we put forward the following suggestions for maintaining the scale of agricultural production: first, we should continue to do a good job in the construction of irrigation and water conservancy projects, and promote the development of water-saving agriculture; second, we should promote the pace of new urbanization, but in the central region we must be reasonable in determining the level of urbanization, because the central region is the main grain production area in China; third, we should strengthen the market price protection policy of agricultural products, raise the price level of agricultural products, and stimulate the enthusiasm of agricultural practitioners to engage in agricultural production.

From the national and regional level, this study reveals the influence of economic development level, agricultural water, the level of urbanization, and agricultural products market price on agricultural production scale, but at the micro level, the benefits of agricultural production determine the practitioners of the scale of production. We now, therefore, want to expand the research of agricultural income effects on agricultural production scale.

## Figures and Tables

**Figure 1 ijerph-18-09085-f001:**
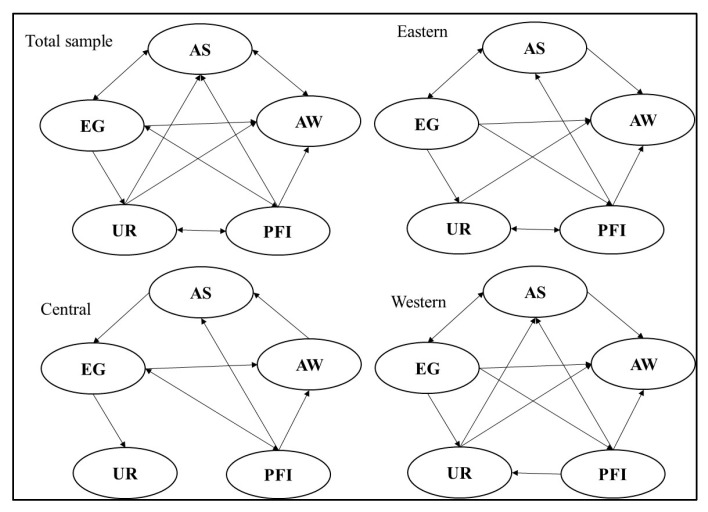
Granger causality relationship flows.

**Table 1 ijerph-18-09085-t001:** Results of Cross-Section Dependence Test.

Variable	Breusch–Pagan LM	Pesaran Scaled LM	Bias-Corrected Scaled LM	Pesaran CD
AS	2686.866 ***	76.34539 ***	75.40789 ***	7.775432 ***
EG	7127.791 ***	226.9069 ***	225.9694 ***	84.40667 ***
AW	2184.176 ***	59.30261 ***	58.36511 ***	2.601526 ***
UR	6529.159 ***	206.6113 ***	205.6738 ***	80.10439 ***
PFI	7163.443 ***	228.1156 ***	227.1781 ***	84.63157 ***

Note: *** denotes the significance level at 1%.

**Table 2 ijerph-18-09085-t002:** Results of panel unit root test.

	Variable	lnAS		lnEG		lnAW		lnUR		lnPFI	
		Level	First Difference	Level	First Difference	Level	First Difference	Level	First Difference	Level	First Difference
Total sample	Levin–Lin–Chu t *	−3.668 ***	−19.380 ***	0.891	−14.316 ***	−2.093 **	−15.574 ***	−7.107 ***	−20.784 ***	−0.405	−18.199 ***
	Im–Pesaran–Shin W-stat	−0.420	−13.057 ***	8.794	−8.436 ***	0.097	−13.072 ***	−2.452 ***	−14.836 ***	6.094	−13.756 ***
Eastern	Levin–Lin–Chu t *	−2.441 ***	−12.549 ***	0.188	−8.823 ***	−2.012 **	−8.068 ***	−6.197 ***	−10.251 ***	−0.623	−10.207 ***
	Im–Pesaran–Shin W-stat	−0.857	−8.076 ***	4.796	−5.999 ***	−1.201	−7.826 ***	−2.099 **	−7.508 ***	3.809	−7.670 ***
Central	Levin–Lin–Chu t *	−1.513 *	−9.740 ***	0.111	−7.769 ***	0.023	−9.924 ***	−1.898 **	−9.625 ***	−0.673	−9.449 ***
	Im–Pesaran–Shin W-stat	0.370	−6.891 ***	5.254	−4.204 ***	1.014	−7.845 ***	0.047	−7.766 ***	3.092	−7.254 ***
Western	Levin–Lin–Chu t *	−2.571 ***	−11.305 ***	1.489	−8.212 ***	−1.496 *	−9.027 ***	−3.775 ***	−15.588 ***	0.485	−11.648 ***
	Im–Pesaran–Shin W-stat	−0.156	−7.621 ***	5.299	−4.303 ***	0.551	−7.003 ***	−2.142 **	−10.656 ***	3.645	−9.048 ***

**Note:** *** denotes the significance level at 1%, ** denotes the significance level at 5%, * denotes the significance level at 10%.

**Table 3 ijerph-18-09085-t003:** Results of Johansen cointegration test.

Hypothesized: No. of Cointegrating Equations (s)	Total Sample				Eastern				Central				Western			
	Trace Statistic	*p*-Value	Max-Eigen Statistic	*p*-Value	Trace Statistic	*p*-Value	Max-Eigen Statistic	*p*-Value	Trace Statistic	*p*-Value	Max-Eigen Statistic	*p*-Value	Trace Statistic	*p*-Value	Max-Eigen Statistic	*p*-Value
None	454.354	0.000	234.314	0.000	189.144	0.000	106.424	0.000	158.420	0.000	74.421	0.000	197.784	0.000	88.150	0.000
At most 1	220.040	0.000	153.637	0.000	82.720	0.003	41.230	0.006	83.999	0.002	39.122	0.011	109.633	0.000	52.047	0.000
At most 2	66.403	0.000	34.205	0.006	41.491	0.174	22.659	0.189	44.877	0.093	34.809	0.005	57.586	0.005	33.744	0.007
At most 3	32.197	0.026	23.006	0.027	18.832	0.505	14.896	0.296	10.068	0.979	7.214	0.945	23.842	0.207	18.876	0.101
At most 4	9.192	0.348	7.913	0.388	3.936	0.909	3.857	0.874	2.854	0.973	2.341	0.981	4.966	0.812	4.331	0.823
At most 5	1.278	0.258	1.278	0.258	0.079	0.779	0.079	0.779	0.514	0.474	0.514	0.474	0.635	0.425	0.635	0.425

**Table 4 ijerph-18-09085-t004:** Regional long-term regression results used panel FMOLS model.

	Total Sample			Eastern			Central			Western		
Variable	Coefficient	*t*-Statistic	Prob.	Coefficient	*t*-Statistic	Prob.	Coefficient	*t*-Statistic	Prob.	Coefficient	*t*-Statistic	Prob.
lnEG	0.311	1.102	0.271	0.221	0.602	0.548	−1.678	−4.159	0.000	−0.349	−0.689	0.492
lnEG^2^	−0.026	−1.985	0.048	−0.030	−1.837	0.068	0.085	4.473	0.000	0.013	0.538	0.592
lnAW	0.634	11.424	0.000	0.836	13.407	0.000	0.119	1.765	0.080	0.159	1.111	0.269
lnUR	0.377	2.817	0.005	0.613	3.451	0.001	−0.465	−3.616	0.000	0.270	1.021	0.309
lnPFI	0.292	3.459	0.001	0.602	5.192	0.000	0.351	3.978	0.000	0.182	1.340	0.183

**Table 5 ijerph-18-09085-t005:** Regional long-term regression results used fixed-effects model.

	Total Sample			Eastern			Central			Western		
Variable	Coefficient	*t*-Statistic	Prob.	Coefficient	*t*-Statistic	Prob.	Coefficient	*t*-Statistic	Prob.	Coefficient	*t*-Statistic	Prob.
C	3.650	4.311	0.000	3.658	2.482	0.014	13.667	10.502	0.000	8.357	6.518	0.000
lnEG	0.399	2.393	0.017	0.162	0.592	0.555	−1.451	−5.291	0.000	−0.224	−0.934	0.352
lnEG^2^	−0.028	−3.733	0.000	−0.025	−2.041	0.043	0.075	5.832	0.000	0.007	0.584	0.560
lnAW	0.579	16.358	0.000	0.757	15.033	0.000	0.129	2.825	0.005	0.122	1.551	0.123
lnUR	0.368	4.321	0.000	0.650	4.551	0.000	−0.408	−4.549	0.000	0.256	1.886	0.061
lnPFI	0.219	3.996	0.000	0.451	4.722	0.000	0.262	4.088	0.000	0.189	2.624	0.010

**Table 6 ijerph-18-09085-t006:** Results of Pairwise Dumitrescu–Hurlin Panel Causality Tests.

Null Hypothesis:	Total Sample		Eastern		Central		Western	
	Zbar-Stat.	Prob.	Zbar-Stat.	Prob.	Zbar-Stat.	Prob.	Zbar-Stat.	Prob.
lnEG does not homogeneously cause lnAS	4.773	0.000	2.125	0.034	1.324	0.185	4.936	0.000
lnAS does not homogeneously cause lnEG	5.521	0.000	2.009	0.045	3.784	0.000	3.977	0.000
lnAW does not homogeneously cause lnAS	2.753	0.006	1.536	0.125	1.981	0.048	1.273	0.203
lnAS does not homogeneously cause lnAW	3.842	0.000	4.522	0.000	−0.030	0.976	1.822	0.068
lnPFI does not homogeneously cause lnAS	7.742	0.000	3.764	0.000	1.881	0.060	7.907	0.000
lnAS does not homogeneously cause lnPFI	0.232	0.816	0.254	0.800	−0.034	0.973	0.165	0.869
lnUR does not homogeneously cause lnAS	2.210	0.027	0.086	0.932	1.006	0.314	2.931	0.003
lnAS does not homogeneously cause lnUR	−0.393	0.694	−1.293	0.196	−0.408	0.683	1.183	0.237
lnAW does not homogeneously cause lnEG	1.346	0.178	0.388	0.698	0.362	0.717	1.648	0.099
lnEG does not homogeneously cause lnAW	9.365	0.000	6.319	0.000	4.008	0.000	5.794	0.000
lnPFI does not homogeneously cause lnEG	2.449	0.014	1.259	0.208	2.468	0.014	0.550	0.583
lnEG does not homogeneously cause lnPFI	8.727	0.000	5.742	0.000	3.156	0.002	6.147	0.000
lnUR does not homogeneously cause lnEG	−0.219	0.826	0.488	0.625	−0.033	0.973	−0.931	0.352
lnEG does not homogeneously cause lnUR	10.204	0.000	3.628	0.000	2.568	0.010	11.873	0.000
lnPFI does not homogeneously cause lnAW	9.854	0.000	6.129	0.000	6.419	0.000	4.495	0.000
lnAW does not homogeneously cause lnPFI	−0.879	0.380	−0.725	0.469	−1.153	0.249	0.385	0.700
lnUR does not homogeneously cause lnAW	7.794	0.000	6.377	0.000	0.765	0.444	6.101	0.000
lnAW does not homogeneously cause lnUR	1.615	0.106	0.309	0.758	−1.052	0.293	3.643	0.000
lnUR does not homogeneously cause lnPFI	1.924	0.054	2.311	0.021	−0.222	0.824	1.068	0.286
lnPFI does not homogeneously cause lnUR	2.500	0.012	1.723	0.085	−0.667	0.505	3.240	0.001

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
