# Peer review of "Impact of Economic Development Level and Agricultural Water Use on Agricultural Production Scale in China"

_ijerph, 2021, doi:10.3390/ijerph18179085_

Round 1

Reviewer 1 Report

Dear author,

Please make some minor spell check required. 

Author Response

Dear Reviewer,

First of and foremost, we would like to thank you for the opportunity to revise our manuscript. Thanks a lot for your praising words, as well as your careful criticism of our work. The comments are very clear, detailed, fair and helpful, which has helped us to make major improvements to our manuscript.

We went over the manuscript again and corrected spelling and grammatical errors.

Best regards,

Jiaxing Pang

Reviewer 2 Report

The manuscript “Impact of economic development level and agricultural water use on agricultural production scale in China” offers a complex analysis on the relationships between economic development, water use, and agricultural productivity in China. Overall, the analyses are excellent, well explained, and justified. However, this manuscript needs a good English revision and editing, as many parts are hard to read. Please find below several specific comments. Abstract L.13. Change “urbanization and the” for “urbanization, and the”. L.14. Why in China specifically? L.17. Same as on L.13. L.20. Change “0.377% and 0.292%” for “0.377%, and 0.292%”. L.26. Change “water use and the” for “water use, and the”. L.27. Change “all of variables” for “all variables”. L.28-32. This phrase needs to be better written. Introduction L.47. Change “cannot delayed” for “cannot be delayed”. L.48-49. Give references and/or examples of this. L.54. Change “its population” for “although its population”. L.57. Change “China facing” for “China is facing”. L.53-57. But what about productivity per hectare? I understand the cultivated area (4th in the world) is way too low compared to the big population of China (1st in the world), but what about productivity per area? Is high, low, medium? It compensates, at least in some way, the low cultivated area? You should also mention this. L.63-64. Non-agriculture what? Be specific. L.66. Change “threat, the” for “threat, and the”. L.85. “China's mountainous” what? L.89. Change “in the southern” for “in southern”. L.99. Change “are interrelated and interrelated” for “are interrelated”. L.104-105. This phrase is not clear. L.109. Change “price and other” for “price, and other”. L.110. Avoid saying things like “this study studies”. L.111. Change “price and urbanization” for “price, and urbanization”. Data and Methodology L.117. Change “Hong Kong and Macao” for “Hong Kong, and Macao”. L.119. Change “(2004–2020) and” for “(2004–2020), and”. L.128. Change “central and western” for “central, and western”. L.130. Change “Guangxi and Hainan” for “Guangxi, and Hainan”. L.131. Change “Hubei and Hunan” for “Hubei, and Hunan”. L.132. Change “Ningxia and Xinjiang” for “Ningxia, and Xinjiang”. L.135. Change “consumption and sown” for “consumption, and sown”. Add a comma (,) after “central”. L.166. Change “The Fully FMOLS” for “The FMOLS”. L.177. Change “formula is follows” for “formula is as it follows”. Results and Discussion L.183. Change “test is obtained” for “test were obtained”. L.185. Change “LM and Pesaran” for “LM, and Pesaran”. L.187. Change “It implies” for “This implies”. L.189. Change “can be influence” for “can influence”. L.200. Change “UR and PFI” for “UR, and PFI”. L.202. Same as in L.200. L.206-207. This phrase is not clear. L.218. Change “consumption have an” for “consumption has an”. L.225. Change “with Kuznets” for “with the Kuznets”. L.228. Change “rate and agricultural” for “rate, and agricultural”. L.230. Change “0.377 and 0.292” for “0.377, and 0.292”. L.237. Change “rate and agricultural” for “rate, and agricultural”. L.239. Change “0.613 and 0.602” for “0.613, and 0.602”. Change “In the central region and western region” for “In the central and western regions”. L.273. Change “worker do not reach” for “worker does not reach”. L.285-286. “that the existence of cointegration between variables” what? L.306. Change “there are” for “there is”. Conclusions L.315. Change “economy and the” for “economy, and the”. L.316. Change “better researching of the effects” for “better research on the effects”. L.317. Change “urbanization and the” for “urbanization, and the”. L.319-322. This is redundant, delete it. L.328. Change “urbanization and the” for “urbanization, and the”. L.331. Change “0.377% and 0.292%” for “0.377%, and 0.292%”. L.343. Change “0.119% and 0.351%” for “0.119%, and 0.351%”. L.351. Change “There are” for “There is”.

Reviewer 3 Report

See attached comments and suggestions for the authors 

Reviewer 4 Report

This article covers an important topic of agricultural production in a country undergoing major economic and socio-economic changes. A thorough analysis of the impact of several key factors on agricultural production size may support estimation of the risk related to the country's food security in the future.

I recommend this work to be published after a minor revision.

General comments:

Abstract

The abstract is too long and far exceeds the MDPI standards. Please keep only the most important information and shorten the abstract to 200 words.

Introduction

The introduction part is sufficient to define the background of the research. It also allows to recognise that the examined issue is an important problem. However, the author should consider placing at least 4-5 cases more researching this issue in other regions of the world.

Data and Methodology

Data and methods are described well enough, with appropriate references.

Results and Discussion

The paragraph contains results but there is no discussion. The discussion should refer to other studies and compare them with the results obtained by the authors.

Discussion

Please add a discussion part to the text. This section could also be called a summary and discussion, summarizing the results obtained and relating them to others on a similar subject.

Below, I propose some papers that the authors can use. I also encourage you to look for others.

1)Urbanization and its implications for food and farming David Satterthwaite, Gordon McGranahan, Cecilia Tacoli Philos Trans R Soc Lond B Biol Sci. 2010 Sep 27; 365(1554): 2809–2820. doi: 10.1098/rstb.2010.0136

2)The estimation of non-irrigated crop area and production using the regression analysis approach: A case study of Bursa Region (Turkey) in the mid-nineteenth century Eda Ustaoglu, M. Erdem Kabadayı, Petrus Johannes Gerrits PLoS One. 2021; 16(4): e0251091. Published online 2021 Apr 30. doi: 10.1371/journal.pone.0251091

 3) Fitton N, Alexander P, Arnell N, et al.. The vulnerabilities of agricultural land and food production to future water scarcity. Glob Environ Chan. 2019; 58: 101944

4) Ustaoglu E, Collier M. Farmland abandonment in Europe: An overview of drivers, consequences and assessment of the sustainability implications. Environ Rev. 2018; 26(4): 396–416

5) Krupnik TJ, Schulthess U, Ahmed ZU, McDonald AJ. Sustainable crop intensification through surface water irrigation in Bangladesh? A geospatial assessment of landscape-scale production potential. Land use policy. 2017 Jan;60:206-222. doi: 10.1016/j.landusepol.2016.10.001. PMID: 28050058; PMCID: PMC5142720.

Conclusions

This part needs to be rebuilt. There are too much information which are not conclusions. For example, lines 314-325 are more appropriate for the "Summary and Discussion" section (and you can build up on that the proposed discussion section).

Please choose the most important content from the rest of the section. Consider including this as a list.

Author Response

Dear Reviewer,

First of and foremost, we would like to thank you for the opportunity to revise our manuscript. Thanks a lot for your praising words, as well as your careful criticism of our work. The comments are very clear, detailed, fair and helpful, which has helped us to make improvements to our manuscript.

We have revised the manuscript according to your comments and suggestions and presented it in revised form for your review.

Best regards,

Jiaxing Pang

This manuscript is a resubmission of an earlier submission. The following is a list of the peer review reports and author responses from that submission.